# Quantify the impacts of climate change and human agricultural activities on oasis water requirements in an arid region: A case study of the Heihe River Basin, China

Xingran Liu[1,2], Yanjun Shen[1]

[1]Key Laboratory of Agricultural Water Resources, Hebei Laboratory of Agricultural Water-Saving, Center for Agricultural Resources Research, Institute of Genetics and Developmental Biology, Chinese Academy of Sciences, 286 Huaizhong Road, Shijiazhuang 050021, China;

[2]University of Chinese Academy of Sciences, Beijing 100049, China

*Correspondence to*: Yanjun Shen (yjshen@sjziam.ac.cn)

**Abstract**. Ecological deterioration in arid regions caused by agricultural development has become a global issue. Understanding water requirements of the oasis ecosystems and the influences of human agricultural activities and climate change is important for the sustainable development of oasis ecosystems and water resources management in arid regions. In this study, water requirements of the main oasis in Heihe River Basin during 1986-2013 were analyzed and the amount showed a sharp increase from $10.8 \times 10^8$ m$^3$ in 1986 to $19.0 \times 10^8$ m$^3$ in 2013. Both human agricultural activities and climate change could lead to the increase of water requirement amount. To quantify the contributions of agricultural activities and climate change to the increase in water requirements, partial derivative and slope method were used. Results showed that climate change and human agricultural activities, such as oasis expansion and changes in land cropping structure, has contributed to the increase of water requirement at rates of 6.9%, 58.3%, and 25.4%, respectively. Overall, human agricultural activities were the dominant driving forces for the increase of water requirement amount. And the contribution of oasis expanding to the increased water requirement amount was significantly greater than that of other concerned variables. This reveals that to control the oasis scale is extremely important and effective to balancing water for agriculture and ecosystems and to achieving a sustainable oasis development in arid regions.

## 1. Introduction

Inland river basins take up about 11.4 % of the land area in the world and most of them are distributed over arid regions (Li et al., 2013). Water resources in arid regions are scarce and critical to ecosystems and societies. For the inland river basins in arid regions, water resources mainly originate from the precipitation and snow/glacier melting in the upstream mountainous areas, and are consumed mainly by agriculture and human

society in oases of the piedmont plains in the midstream, then finally are discharged and dispersed in the tail lakes in the downstream (Kang et al., 1999; Shen and Chen, 2010). The precipitation in plain areas or in major economic centers of arid basins has nearly no significant meaning for generating runoff (Shen and Chen, 2010).

Owing to scarce water resources in arid regions, ecosystems and societies are vulnerable to hydrologic changes. With the rapid growth of population in arid regions of the world (Shen and Chen, 2010), the utilization of surface- and groundwater for irrigation increased without enough consideration for ecological conservation, which caused severe deterioration of water and ecosystems in most arid river basins. For example, the Amu Darya and the Syr Darya are two main rivers in the Central Asia, which flow towards the Aral Sea. More than 90% of the water withdrawal in the region was used for agricultural irrigation (Sorg et al., 2014). With the increase in irrigated area in the past decades, irrigation withdrawals have measurably reduced inflow to the Aral Sea since 1960s, which caused significant shrinking of the water surface of the Aral Sea and land desertification, and even the fishery in the Aral Sea has almost been destroyed because of salinization (Micklin, 1988; Sorg et al., 2014; Shen and Chen, 2010; Karimov and Matthies, 2013). Similarly, the disappearance of Lop Nor in western China, the dying of the Dead Sea in the Middle East, and the shrinking of Lake Chad in Africa are all notable examples. Ecological deterioration in arid regions caused by agricultural development has become a global issue and has become the main obstacle to the sustainable development of oasis ecosystems.

Despite human exploitation, climate change can also influence the water resources in arid regions. It is reported that the climate in arid regions has become drier in the past century (Narisma et al., 2007; Dai et al., 2004), showing increasing temperature, variability of precipitation, and reduction of glaciers and snow areas (Wang and Qin, 2017) and would be more arid in the future (Bates et al., 2008). But huge amount of studies suggested that the water and ecological degradation in arid regions was largely affected much more by irrational human exploitation than climate change (Jarsjö et al., 2008; Aus der Beek et al., 2011; Huo et al., 2008; Dong et al., 2014; Ma et al., 2014).

The ecological degradation and water shortages have heightened the importance of water allocated to the agriculture in the oasis ecosystems. Water requirement is an important parameter for irrigation scheduling and regional water allocation. Studies on water requirements are theoretically and practically indispensable for the sustainable development of oasis ecosystems in arid regions. Scientists have obtained some research results about water requirements of oasis ecosystems, including the crop water requirements (Kawy and El-Magd, 2012; Liu et al., 2010; Siebert et al., 2007; Zhao et al., 2005; Zhao et al., 2010; De Silva et al., 2007; Kawy and Darwish, 2013), and ecological water requirements (Guo et al., 2016; Ye et al., 2010; Zhao et al., 2007; Guo et al., 2016). Studies have shown that the water requirement would increase if the climate becomes drier and warmer (Döll, 2002; Nkomozepi and Chung, 2012; Fu et al., 2014), and human activities have gradually became the predominant factor increasing the water requirement amount in the past decades (Bai et al., 2014; Coe and Foley, 2001; Zou et al., 2017). But there are few studies separately quantify the contributions of climate change and human agricultural activities to changes in water requirement

amount.
Approximately one quarter of land area in China located in arid regions. As the second
largest inland river in China, Heihe River Basin also suffered water conflict between
agricultural development and ecological health and was chosen as the target basin for a
key national research programme on ecohydrology and integrated basin water
management by the Natural Scientific Foundation of China in 2012, and the programme
is still going on. So the oasis in the middle Heihe River Basin where more than 90% of
the arable land were concentrated was taken as the study area. The main objectives of
this study are to make clear the changes in water requirements in the oasis under climate
change and human agricultural activities and identify the main factor that influences
the changes in water requirement amount based on the clarification on the contributions
of climate and human activities, including land structure and area, to the changes in
water requirement amount. The research questions addressed were: (1) How have the
water requirements of the oasis changed in the past ~30 years? (2) Why the water
requirement amount of the oasis have changed? We anticipate that this study would be
valuable as a reference to the water resources research for the global arid regions.

**2. Material and methods**
**2.1 Study area**
Heihe River originates in the Qilian Mountains, and flows to the oases in the piedmont
plain after reaching the mountain outlet at the Yingluo Gorge, then finally terminates at
the East and West Juyan Lakes. It breeds an ecosystem which consists of ice-snow,
frozen soil, and mountain vegetation zones at the upstream, and oasis zone and desert
zone at the middle and down streams (Ersi et al., 1999; Kang et al., 2005; Zhao et al.,
2007). The study was conducted in the oasis in the middle Heihe River Basin (between
$38°\ 32'$ and $39°\ 52'$ N, and $98°\ 57'$ and $100°\ 51'$ E), China (Fig. 1). It embraces
a total area of $8.6\times10^9$ m$^2$, included in Ganzhou district, Linze county and Gaotai county.
More than 90 % of the population and arable land in the Heihe River Basin were
concentrated in this oasis (Zhang et al., 2006).
Situated in the inner of Asia-Europe continent, the study area possesses a temperate
continental arid climate with sufficient sunlight, great temperature variations and scarce
precipitation. According to the observed data by Gaotai and Zhangye meteorological
stations in the study region during 1953-2014, the annual average temperature is about
6.0-9.4 °C, with the lowest temperatures occurring in January and December, and
highest temperatures occurring in July. The annual sunshine in the region is about 2800-
3400 h. The mean annual precipitation is less than 130 mm (e.g. the mean annual
precipitation is 107.86 mm and 129.10 mm at Gaotai and Zhangye meteorological
stations, respectively). Over 60 % of the precipitation falls between June and August
(Zhao et al., 2005). But the annual potential evaporation reaches 1400 mm (Li et al.,

2016).

The study area has an agricultural development history of over 2000 years owing to its flat land, adequate sunlight, and convenient water resource from Qilian Mountains. The oasis in the middle Heihe River Basin has then become an important commodity grain base in China. Combined with the cultivated land, forest, grass, swampland, and waters make up the oasis together. The oasis area has been expanding in the recent ~30 years. According to the land use data developed by the Chinese Academy of Sciences (CAS), the oasis area increased ~ 906 km$^2$ during the past decades, in which the cultivated land increased about 740 km$^2$ (Fig. 2). And the cropping pattern has also changed a lot in the recent ~30 years (Fig. 3). The area of maize increased significantly; on the contrary, the wheat planting area decreased evidently. Besides, the planting area of vegetable also increased especially in Gaotai county during the past ~30 years. The cropping pattern in the study area are turning to be simple to focus on the maize, which providing more than 40 % of maize seeds in China (Xing, 2013).

Lacking in precipitation, surface runoff has become the main surface water resources for irrigation. The middle Heihe River flows from Yingluo Gorge to Zhengyi Gorge, supplying water for oasis in the middle river basin. Annual discharge observed at Yingluo Gorge increased from around 14.4×10$^8$ m$^3$ in the 1960s to 15.7×10$^8$ m$^3$ in the 1990s, while the discharge observed at Zhengyi Gorge decreased from around 10.5×10$^8$ m$^3$ in the 1960s to around 7.5×10$^8$ m$^3$ in the 1990s (Wang et al., 2014). The development of modern irrigation schemes, and the growth of population and irrigation area in the middle basin took up an increasing share of water resources, endangering the hydrological conditions, ecology and environment in the Heihe River Basin (Chen et al., 2005; Jia et al., 2011;). More than 30 tributaries as well as the terminal lakes have dried up, and the discharge in the downstream decreased significantly in the past 50 years (Wang and Cheng, 1999; Chen et al., 2005). Such hydrological changes have resulted in a marked degradation of the ecological environment, land salinization and desertification in the entire basin. To restore the ecosystem of the downstream, the Ecological Water Diversion Project (EWDP) was launched by the Chinese Government around the year 2000, which stipulated that water flowing from the Zhengyi Gorge to the downstream should be over 9.5×10$^8$ m$^3$ when the annual average water supplied from the Yingluo Gorge is 15.8×10$^8$ m$^3$ (Zhao et al., 2016). Due to the EWCP, the discharge observed at Zhengyi Gorge has increased since 2000, which led to less available surface water for the middle Heihe River Basin, and more groundwater was taken for irrigation (Ji et al., 2006). According to the groundwater withdrawal data (1990-2007) of the irrigation districts in the middle Heihe River Basin (Wu et al., 2014; Zheng, 2014), which were downloaded from the Cold and Arid Regions Science Data Center at Lanzhou (http://westdc.westgis.ac.cn), only 1.13×10$^8$ m$^3$ of groundwater was pumped on average in 1990s, but the amount increased to 3.25×10$^8$ m$^3$ on average during 2000-2007.

## 2.2 Data handling and processing

### 2.2.1 Meteorological data

Daily meteorological observations were collected from China Meteorological Administration (CMA), mainly including the maximum, minimum and average air temperatures, wind speed, relative humidity and sunshine duration. 10 meteorological stations, which covered the Gaotai, Zhangye stations inside the study region and Dingxin, Jinta, Jiuquan, Tuole, Yeniugou, Qilian, Shandan, Alxa youqi stations outside the study region, were selected to get the spatial distribution of meteorological elements (Fig. 1). Observations on crop growth and phenology were collected from the agricultural meteorological stations in Gansu Province, especially from the station in Zhangye. But the data on crop growth and phenology were only basically recorded completely for the maize (1993-2013) and spring wheat (1992-2013), so the growth and phenology data for other vegetation were obtained by references (Liu, 2014; Allen et al., 1998; Pu et al., 2004; Li et al., 2009; Zhou et al., 2015 ), combining practical investigation. The growth and phenology data for maize before 1993 were set as that in 1993, and for spring wheat before 1992 were set as that in 1992.

### 2.2.2 Land use data

Land use data for years 1986, 1995, 2000, 2011 at a spatial resolution of 30 m (Wang et al., 2011a, b; Liu et al., 2003; Wang et al., 2014), which were developed by CAS, were used in this study. The same classifying system for land cover was applied to the four years' land use data. The land-use patterns in the basin have been divided into 6 types: cultivated land; forest land which includes closed forest land, sparse wood land, shrubs, and other wood land; grassland which contains high coverage grassland, moderate coverage grassland, and low coverage grassland; waters which comprise rivers, lakes, reservoirs, and beach land; construction land; and unused land which contains sand, gobi, saline-alkali land, swampland, bare land, bare rock and gravel. To get the continuous land use maps, the land use data at the spatial resolution of 30 m were transformed to the land use data at the spatial resolution of 1 km in the form of percentage. Then the spatial distribution of the land use data between the four discrete years could be obtained by linear interpolation.

To obtain the spatial distribution of specific crops in the cultivated land, the socio-economic statistical data were collected from the *Gansu Development Yearbook* (1984-2014) and *Gansu Rural Yearbook* (1990-2014), including various crops sown at the county level. Based on the main crops in the Statistical Yearbooks, the cultivated land was classified into 7 types: maize, spring wheat, cotton, oilseed, sugar beet, potato, and vegetable. According to the proportion of each crop in each county (Fig. 3), the spatial distribution of the seven crops were determined.

 **2.2.3 Validation data**

The water requirements estimated in this study were compared with two
Evapotranspiration (ET) datasets provided by Cold and Arid Regions Science Data
Center at Lanzhou (http://westdc.westgis.ac.cn). One was the monthly ET datasets
(2000-2013) at 30 m spatial resolution (Wu et al., 2012; Liu et al., 2011) estimated by
ETWatch developed by Wu et al. (2008) and Xiong et al. (2010) for monitoring spatial
ET for operational purposes, while this datasets only covered part of the oasis which
included Ganzhou district, Linze county and small part of the Gaotai county in the
middle Heihe River Basin. The other was the daily ET datasets (2009-2011) of the
Heihe River Basin at 1 km spatial resolution (Cui and Jia, 2014; Jia et al., 2013)
estimated by ETMonitor, which is a hybrid remotely sensed actual ET estimation model
developed by Hu and Jia (2015). The intersections of the ET datasets and water
requirements were used for comparison.
**2.3 Estimates of water requirements**
In this study, the water requirements of the cultivated land, forest land, high coverage
grassland, moderate coverage grassland, waters except the beach land, and the
swampland in the unused land were considered. The water requirements of the low
coverage grassland, beach land, construction land, and unused land except the
swampland were taken as zero.
**2.3.1 Water requirements for the cultivated land and grassland**
Water requirements of the crops and grass in the oasis refer to the evapotranspiration
from disease-free, well-fertilized crops, grown in large fields, under optimum soil water
conditions and achieving full production under the given climatic conditions. This can
be calculated using crop coefficient approach as following:
$$ET_c = K_c \times ET_0 \tag{1}$$
where $ET_c$ is the water requirement; $K_c$ is the crop coefficient; $ET_0$ is the reference
evapotranspiration.
$ET_0$ was calculated using the modified Penman-Monteith equation recommended by
United Nations Food and Agriculture Organization (FAO) (Allen et al., 1998).
Reference evapotranspiration is only related to meteorological factors (Shahid, 2010).
It can be used in a wide range of locations and climates, and can be calculated using the
following equation:
$$ET_0 = \frac{0.408\Delta(R_n - G) + \gamma(900/(T+273))u_2(e_s - e_a)}{\Delta + \gamma(1 + 0.34u_2)} \tag{2}$$
where $ET_0$ is the reference evapotranspiration (mm); $R_n$ is the net radiation at crop
surface [MJ/(m$^2$ d)]; $G$ is the soil heat flux density [MJ/(m$^2$ d)]; $u_2$ is the wind speed at
a height of 2 m (m/s); $T$ is the mean daily air temperature at a height of 2 m (℃); $e_s$ is
the saturation vapor pressure (kPa); $e_a$ is the actual vapor pressure (kPa); $\Delta$ is the slope
of the vapor pressure-temperature curve [kPa/℃]; and $\gamma$ is the psychrometric constant
[kPa/℃].
Different vegetation types have different $K_c$ coefficients. The changing characteristics
of the vegetation over the growing season also affect the $K_c$ coefficient, so $K_c$ for a given
vegetation type will vary over the growing period, which can be divided into four
distinct growth stages: initial, crop development, mid-season and late season. In the
current study, $K_c$ for different crop species in the cultivated land during the four growth
stages were determined according to Duan et al. (2004) and FAO (Allen et al, 1998).
And $K_c$ for the grassland were determined according to Liu (2014). The $K_c$ values are
shown in Table 1.

### 2.3.2 Water requirements for the forest land

For the forest land, the water requirements of closed forest land, sparse wood land and
shrubs were estimated by phreatic evaporation. It can be calculated as below:

$$W_i = S_i \times W_{gi} \times k_p \tag{3}$$

where $W_i$ is the ecological water demand of vegetation $i$; $S_i$ is the area of vegetation
type $i$; $W_{gi}$ is the phreatic evaporation capacity of the vegetation type $i$ at a certain
groundwater depth; $k_p$ is the vegetation coefficient, which is related to the groundwater
depth (Table 2) (Song et al., 2000).
$W_{gi}$ is the key to calculate vegetation ecological water demand using the phreatic
evaporation method, and it is usually calculated using Averyanov's phreatic evaporation
equations:

$$W_{gi} = a(1 - h_i/h_{max})^b E_0 \tag{4}$$

where a and b are empirical coefficients (0.856 and 3.674 in the study area) (Wang and
Cheng, 2002); $h_i$ is the groundwater depth of vegetation type $i$, which is 1.5 m, 2 m, 2.5
m for the closed forest land, sparse wood land and shrubs, respectively; $h_{max}$ is the
maximum depth of phreatic evaporation, which is 5 m (Wang and Cheng, 2002); and
$E_0$ is the surface water evaporation.
The other wood in the study area was mainly orchard, so the water requirement of other
wood land was calculated by the crop coefficient approach (Table 1).

### 2.3.3 Water requirements for waters and the swampland

The water requirement of waters can be taken as the evaporation from water surfaces,
which can be calculated according to Shuttleworth (1993):

$$ET_w = \frac{\Delta R_n + 6.43\gamma(1+0.536u_2)(e_s - e_a)}{\lambda(\Delta + \gamma)} \tag{5}$$

where $ET_w$ is the water requirement of waters (mm); $\lambda$ is the latent heat of vaporization
(MJ kg$^{-1}$).
The water requirement for the swampland was calculated by crop coefficient approach.
The $K_c$ values of the vegetation in the swampland were determined depending on the
single crop coefficients suggested in FAO (Table 1).
**2.4 Contribution assessment**
According to the methods to estimate water requirements of the oasis in the middle
Heihe River Basin, the value of the water requirements ($y$) is mainly related to the
climate ($x_1$), total area of the oasis ($x_2$), and area proportions of the land structure ($x_3$).
Mathematically, the function can be write as
$$y = f(x_1, x_2, x_3, \dots) \tag{6}$$

The variation of the dependent variable $y$ can be expressed by a differential equation as
$$dy = \frac{\partial f}{\partial x_1} dx_1 + \frac{\partial f}{\partial x_2} dx_2 + \frac{\partial f}{\partial x_3} dx_3 + \cdots \tag{7}$$

As $y$ varies with time $t$, we can rewrite Eq. (7) as
$$\frac{dy}{dt} = \frac{\partial f}{\partial x_1}\frac{dx_1}{dt} + \frac{\partial f}{\partial x_2}\frac{dx_2}{dt} + \frac{\partial f}{\partial x_3}\frac{dx_3}{dt} + \delta \tag{8}$$

$\frac{dy}{dt}$ is the slope of the linear regression for $y$ against time $t$; $\frac{\partial f}{\partial x_1}\frac{dx_1}{dt}$ can be taken as the
slope of the linear regression for $y$ against time $t$ when $x_2$ and $x_3$ don't change with the
time; $\frac{\partial f}{\partial x_2}\frac{dx_2}{dt}$ can be taken as the slope of the linear regression for $y$ against time $t$ when
$x_1$ and $x_3$ don't change with the time; $\frac{\partial f}{\partial x_3}\frac{dx_3}{dt}$ can be taken as the slope of the linear
regression for $y$ against time $t$ when $x_1$ and $x_2$ don't change with the time. Because the
spatial distribution of the climate is not homogeneous, the location where a certain land-
use type is located can also affect the water requirement. So other factors related to the
water requirements were fitted into $\delta$, combining the systemic error.
The individual proportional contribution ($\rho$) of related factors to the long-term trend in
$y$ can be estimated as
$$\rho(x_i) = \left(\frac{\partial f}{\partial x_i}\frac{dx_i}{dt}\right) / \left(\frac{d_y}{d_t}\right) \times 100\% \tag{9}$$

where $x_i$ can be the variable $x_1$, $x_2$ and $x_3$.
**3. Results**
There are 15 specific land-use types in the oasis of the middle Heihe River Basin, which
are cultivated land (maize, spring wheat, cotton, oilseed, sugar beet, potato, vegetable),
grassland (high coverage grassland, moderate coverage grassland), forest land (closed
forest land, sparse wood land, shrubs, the other wood), waters, and swampland.
Different land-use types may require different water amounts. To understand the water
requirements in the oasis, the spatial and temporal variations of the total water
requirement amount and the water requirement per unit area were analyzed. In the study,
the water requirement per unit area for each land-use type were calculated by dividing
the total water requirement of each land-use type by the corresponding land area. After
validation to ensure the accuracy of the results, the water balance and determinants to
the variation of the water requirement amount of the oasis in the middle Heihe River
Basin were analyzed.

### 3.1 Temporal and spatial variations in water requirements of the oasis in the middle Heihe River Basin

The water requirement amount of the total oasis increased from $10.8 \times 10^8 \, m^3$ in 1986 to
$19.0 \times 10^8 \, m^3$ in 2013 (Fig. 4a). According to the land use data, the area of the cultivated
land accounted for ~80 % of the total area of the oasis (Fig. 2). Therefore, the water
requirement amount of the cultivated land increased from $8.4 \times 10^8 \, m^3$ in 1986 to
$14.7 \times 10^8 \, m^3$ in 2013 (Fig. 4a), which occupied 76 %-82 % of the total oasis water
requirement amount during 1986-2013. The mean annual water requirements amount
of the cultivated land and the whole oasis were $10.4 \times 10^8$ and $13.3 \times 10^8 \, m^3$, respectively.
The water requirement amounts of the swampland and waters from 2000 to 2013
increased a lot, so was the water requirement amount of the forest land from 1986 to
1995. But the waters, swampland, forest land, and grassland needed less water amounts
which were all smaller than $1.7 \times 10^8 \, m^3$ because the proportion of them in the oasis were
all smaller than 9 % (Fig. 4a; Fig. 2).
The water requirement of the cultivated land per unit area increased from 519.2 mm to
624.9 mm during 1986-2013, while the water requirement of the oasis per unit area
increased from 527.1 mm to 642.0 mm during 1986-2013 (Fig. 4b). The mean annual
water requirements of the cultivated land and the oasis per unit area were 544.6 mm
and 557.4 mm, respectively. Maize, spring wheat, and vegetable are the main crops in
the middle Heihe River Basin. The mean annual water requirements of the maize, spring
wheat, and vegetable per unit area were 570.0 mm, 413.7 mm, and 728.8 mm,
respectively. Waters required the most water per unit area, the mean annual water
requirement of which reached 1323.9 mm. The swampland covered with reeds also
needed a lot of water per unit area, the mean annual water requirement of which could
reach 968.6 mm. Different land surface coverages for grassland and forest land had
different water requirements. The mean annual water requirements of the closed forest
land, sparse wood land, shrubs, the other wood, high coverage grassland, and moderate
coverage grassland per unit area were 477.5 mm, 128.9 mm, 264.0 mm, 705.1 mm,
663.6 mm, and 340.0 mm, respectively.
The oasis in the middle Heihe River Basin was scattered along the rivers. Most of the
water requirement in the oasis was below 500 mm per square kilometer in 1986
considering the mixed pixel and area weight, but with the climate change and human
agricultural activities, the water requirement in large area of the oasis exceed 500 mm
per square kilometer in 2011 (Fig. 5). And the area of high water requirement in the
oasis accorded with the location of the cultivated land (Fig. 5). Besides, the ecological
vegetation in the oasis except the northwest of Gaotai county didn't show significant
increase in water requirement (Fig. 5).
The cultivated land in most area of the oasis expanded during the past ~30 years,
especially in Linze county and the north of Ganzhou district (Fig. 6a). This was in
accordance with the area of water requirement increased in the cultivated land and the
oasis (Fig. 6). The water requirement in the cultivated land increased above 100 mm
per square kilometer in the Linze county and the north of Ganzhou district. The larger
area the cultivated land expanded, the more water required for the cultivated land (Fig.
6b). Only a small part of the cultivated land shrinked in the oasis and caused the
decrease of water requirement in the corresponding cultivated land (Fig. 6). As the
dominant part affecting the water requirement in the oasis, the spatial distribution of
the increased water requirement in the cultivated land was similar with that in the oasis
(Fig. 6b, c). The water requirement in the northwest of Gaotai county increased
obviously due to the increasing area of swampland after the year of 2000.
**3. 2 Validation of the oasis water requirements**
Water requirement is defined as a theoretical value. For the crops, it can be taken as the
potential crop ET. But there was no available data observed or calculated by others for
the potential crop ET, so the actual ET data were adopted to validate the water
requirement in the oasis to see if the results were acceptable.
According to the yearly and monthly ET estimated by ETWatch and ETMonitor, the
total ET was well correlated with the estimated water requirement amount with the
determination coefficient ($R^2$) of 0.91 (Fig. 7), and slope of the linear regression of 1.05
(Fig. 7). Compared with the yearly ET datasets (2000-2013) estimated by ETWatch
with 30m spatial resolution over part of the oasis, the root mean square error (RMSE)
between the ET and water requirement amount for the cultivated land and the oasis
were $0.71 \times 10^8$ m$^3$ and $0.66 \times 10^8$ m$^3$, respectively. Because the water requirement is the
potential ET, the water requirement should not be smaller than the ET. But the yearly
ET included not only the ET during crop growth period, but also the ET from the bare
land after harvesting the crops. While the estimated water requirement for the crops
only included the water requirement during the crop growth period, so most yearly ET
data were larger than the yearly water requirement amounts (Fig. 7). To remove the
influence of the bare land, the monthly ET datasets in May, June, and July were selected
to validate the water requirement because the vegetation including the crops were all in
their growth period in the three months. It showed that the water requirement was highly
correlated with the ET (Fig. 7). And the RMSE for the cultivated land and the oasis
were $0.35 \times 10^8$ m$^3$ and $0.36 \times 10^8$ m$^3$, respectively, which were much smaller than the
yearly RMSE. Most of the monthly water requirement amounts were higher than the
monthly ET data (Fig. 7).
Compared with the ET datasets (2009-2011) estimated by ETMonitor at 1 km spatial
resolution in the middle Heihe River Basin, the yearly and monthly water requirement
amounts were all larger than the corresponding ET data (Fig. 7), and the RMSE for the
monthly data in May, June, and July was $1.27 \times 10^8$ m$^3$. Because the resolution of the ET
datasets estimated by ETMonitor was relatively low, only the results in the oasis were
validated considering the problem of mixed pixels. The yearly estimated water
requirement amounts in 2009, 2000, and 2011 were smaller than the ET data estimated
by ETWatch for the oasis, which was contrary to the results compared with the ET data
estimated by ETMonitor, which showed that the two ET datasets deviated from each
other, and the estimated water requirements were acceptable.

**3. 3 Water balance in the middle Heihe River Basin**

Yingluo Gorge is the divide of the upper and middle Heihe River, and Zhengyi Gorge
is the divide of the middle and lower Heihe River. The two hydrologic stations recorded
the inflow and outflow of the mainstream of the middle Heihe River. So the surface
runoff of the mainstream of the middle Heihe River consumed in the middle Heihe
River Basin can be considered as the difference between Yingluo Gorge and Zhengyi
Gorge. Besides, there are some small rivers also flow into the middle Heihe River Basin,
like Shandan River and Liyuan River. The mean annual runoff of the Liyuan River and
Shandan River is $2.36\times10^8\,m^3$ (Wu and Miao, 2015) and $0.86\times10^8\,m^3$ (Guo et al., 2000),
respectively. According to the runoff data (1986-2010) of Zhengyi Gorge and Yingluo
Gorge, and precipitation data (1986-2010) obtained from the Cold and Arid Regions
Science Data Center at Lanzhou (http://westdc.westgis.ac.cn) (Yang et al., 2015), the
surface water including the precipitation landing on the oasis and the river discharges
of the middle Heihe River, Shandan River and Liyuan River could meet the water
requirement before the year 2004, ignoring the water conveyance loss. But with the
increasing water requirement of the oasis, the water supply from the land surface could
not meet the requirement any more (Fig. 8).
The vegetation in the oasis can be divided into two categories, one is agricultural
vegetation which includes the crops and orchard, and the other is the ecological
vegetation. The precipitation in the middle Heihe River Basin is too little to supply
enough water for the ecological vegetation (Table 3). The ecological vegetation usually
grows around the cultivated land, so they can absorb the water of infiltration. In addition,
the shelter forest often needs irrigation, and the shrubs like tamarix chinensis and
sacsaoul also need groundwater to maintain normal growth. Compared with the
available water resources in 1980s, precipitation had remained little changed in 1990s.
But with the increase of water requirement in 1990s, the runoff consumed in the middle
Heihe River Basin had an obvious rise and more groundwater was pumped for irrigation
(Fig. 8; Table 3). Ignoring the industrial and domestic water taken from the middle
Heihe River, the surface water supply seemed to be sufficient to the water requirement
in the oasis in 1980s and 1990s. While entering the 20th century, the area of arable land
increased fast, and high water-requiring crops (maize and vegetable) had gradually
replaced the low water-requiring crop (wheat) since 2001 (Fig. 3). Therefore, the water
requirement increased a lot in 2000s. With the implementation of EWCP, the available
surface water from middle Heihe River decreased in 2000s. Surface water cannot meet
the water requirement any more, causing more exploiting of groundwater (Table 3).
The middle Heihe River Basin is in severe water shortage of water resources. To reduce
the contradiction of water supply and requirement, the land use including the crop
structure in the middle Heihe River Basin should be carefully planned.

### 3.4 Contributions to the water requirement trend

Both climate change and human agricultural activities can influence the water requirement of the oasis. In this study, the land expansion, which influences the total oasis area, and the land structure, which influences the area proportion of each land-use type in the oasis, were considered for the human agricultural activities. Because the oasis is dominated by the cultivated land, both the contributions of the influencing factors to the changes in water requirement amount of the oasis and of the cultivated land were analyzed. For the cultivated land, the three influencing factors considered to be the climate change, the expansion of the cultivated land, and the crop structure in the cultivated land. The area of the oasis in 1986, 1995, 2000, and 2011 was 2048.96 $km^2$, 2091.13 $km^2$, 2216.97 $km^2$, and 2954.85 $km^2$, respectively, which showed an obvious increase in the recent ~30 years. For the specific land-use types, the area of cultivated land, waters, and swampland in 2011 showed an obvious increase, compared with the area in 1986. The area of the cultivated land was only 1614.32 $km^2$ in 1986, but it increased to 2354.25 $km^2$ in 2011. Besides the land expansion, the increased area of the land-use types with high water requirement like the vegetable, maize, waters, and swampland also increased the water requirement amount of the oasis.

The water requirement amounts of the oasis and cultivated land increased $0.3447 \times 10^8$ $m^3$ and $0.2743 \times 10^8$ $m^3$ per year during 1986-2013, respectively (Fig. 9a). Considering the impact of climate change on the water requirement amount, the land area and the land structure were set stable, and only the climate changed as usual during 1986-2013. In the situation, the water requirement amount increased slowly at the rates of $0.0238 \times 10^8$ $m^3$ and $0.0184 \times 10^8$ $m^3$ per year for the oasis and cultivated land, respectively (Fig. 9b), which revealed that climate change had a positive effect on the increase in water requirement. Based on Eq. (9), the contributions of the climate change to the increase in water requirement amount were 6.9 % and 6.7 % for the oasis and cultivated land, respectively.

Considering the impact of land expansion on the water requirement amount, the climate and the land structure were set stable, and only the total land area changed with time during 1986-2013. In this situation, the water requirement amount increased rapidly at the rates of $0.2008 \times 10^8$ $m^3$ and $0.1661 \times 10^8$ $m^3$ per year for the oasis and cultivated land, respectively (Fig. 9c), which were nearly 9 times faster than the increasing speed caused by climate change. The contributions of land expansion were 58.3 % and 60.6 % to the increase in water requirement amount for the oasis and cultivated land, respectively.

Considering the impact of land structure on the water requirement amount, the climate and total land area were set stable, and only the land structure changed as usual during 1986-2013. In this situation, the water requirement amount increased at the rates of $0.0874 \times 10^8$ $m^3$ and $0.0645 \times 10^8$ $m^3$ per year for the oasis and cultivated land, respectively (Fig. 9d), which were approximately 4 times faster than the increase speed caused by climate change. The contributions of the land structure were 25.4 % and 23.5 % to the water requirement changes for the oasis and cultivated land, respectively.

The three influencing factors explained approximately 91 % of the increase in the water requirement amounts of the oasis and cultivated land during 1986-2013. In the recent

~30 years, human agricultural activities including land expansion and changes in land structure contributed about 84% to the increase in water requirement amount, and the climate change only contributed about 7% to the increase. And land expansion was the dominant factor contributing to the increase in water requirement amount of the oasis.

## 4. Discussion

Based on the land use and meteorological data, the impact of climate change and human agricultural activities, including land expansion and changes in land structure, on the water requirements of the oasis and the cultivated land which is the main part of the oasis in the middle Heihe River Basin were calculated and analyzed. The results suggest that both climate change and human agricultural activities can lead to the increase in water requirement amounts and the contribution of human agricultural activities to the increase was significantly greater than the climate change. And the land expansion was the dominant factor contributing to the increase in water requirement amounts.

Crop water requirement is the ET from disease-free, well-fertilized crops under optimum soil water conditions and achieving full production. There is no available observed crop water requirement to validate the results. Only actual ET data can be obtained. There are 18 field stations in the oasis that all located in Ganzhou district in the middle Heihe River Basin for conducting meteorological observation and flux measurements from around June, 2012. But due to the incomplete daily data and short time series, we used the ET datasets provided by Cold and Arid Regions Science Data Center at Lanzhou (http://westdc.westgis.ac.cn) to validate the results. Compared with other research results, the mean annual water requirement of the main crop (maize), which was 570.0 mm in this study, basically accorded with the result by Liu et al. (2010). And the mean annual water requirements of cultivated land and wheat, which were 544.6 mm and 413.7 mm, respectively in this study, was consistent with the results by Liu et al. (2017).

Crop coefficient is an important parameter to estimate the water requirement, and it is related to many factors, such as the biological characters of crops, cultivation and soil conditions, etc., so the crop coefficients for different crop varieties of the same crop could be different. Some researchers (Nader et al., 2013; Mu, 2005) studied on the crop coefficients affected by different crop varieties, and found that there were differences in every growth stage between different varieties, and the differences were almost less than 0.3. But it's difficult to get the crop coefficients for every specific crop variety because there are too many varieties. Besides, the water requirement is not only related to the crop coefficient, but also related to the crop growth period. Many factors influencing the crop coefficient also have an effect on the growth stages. Like the study by Nader et al. (2013), the water requirement variation was much smaller than the variation of crop coefficients for different varieties. Therefore, though we didn't distinguish the crop coefficients among different varieties, the estimated water requirements in the study were still reliable.

There are many factors influencing the water requirement. This study only analyzed the

major factors which influence it (climate change and human agricultural activities). Climate change including factors for air temperatures, wind speed, relative humidity and sunshine duration, and Human agricultural activities, including the land expansion and changes in land structure totally contribute about 91% to the increase in water requirement amount of the oasis. Other influential factors, such as changes in location of different land types, are difficult to quantify and were not considered in the study. Besides, some driving factors are not independent, and changes in one factor can cause changes in other factors, such as the climate change and changes in crop phenology. So in the contribution analysis, when the climate were set stable, the crop phenology also kept stable, and when the climate changed, the crop phenology varied according to the statistical data.

As an oasis located at ecologically vulnerable areas and dominated by agriculture, the development of agriculture should match up with the climate and ecological capacity. The water amount consumed in the oasis ecosystem concerns the ecological security of the whole basin. To promote the harmonious development among the upstream, midstream and downstream, the water amount consumed in the agricultural oasis must be controlled and a series of water-saving measures should be carry on. Because the oasis area and the land structure are the main reason why the water requirement amount of the oasis increased so fast, additional efforts will be made to determine the appropriate oasis area and crop structure in the oasis.

## 5. Conclusion

Affected by the climate change and human agricultural activities, the water requirement amount of the oasis increased significantly during 1986-2013, which increased from $10.8 \times 10^8$ m$^3$ in 1986 to $19.0 \times 10^8$ m$^3$ in 2013. Cultivated land is the main part of the oasis, the water requirement amount of which increased from $8.4 \times 10^8$ m$^3$ in 1986 to $14.7 \times 10^8$ m$^3$ in 2013. Contribution analysis identified the dominant factors influencing the water requirement amount were the human activities, the contribution of which including the land expansion and changes in land structure to the increase in water requirement amount was about 84%, and the climate change only contributed about 7% to the increase. For the human activities, land expansion contributed most to the increase in water requirement amount, which contributed 58.3 % and 60.6 % for the oasis and cultivated land, respectively. To reduce the water requirement amount and ensure the sustainable development of oasis ecosystems in arid regions dominated by agriculture, it is necessary to further rationalize the scale of the oasis and cultivated land, and optimize the crop structure.

## 6. Data availability

The meteorological data are available at http://data.cma.cn/. The land use data,

validation data, runoff data, and precipitation data used in this study are available at http://westdc.westgis.ac.cn.

**Competing interests.** The authors declare that they have no conflict of interest.

**Acknowledgements.**

This study was supported by the National Natural Science Foundation of China (No. 91425302). We would like to extend our thanks to Yulu Zhang from Qinghai Normal University, who helped process the wind speed data in three meteorological stations.

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

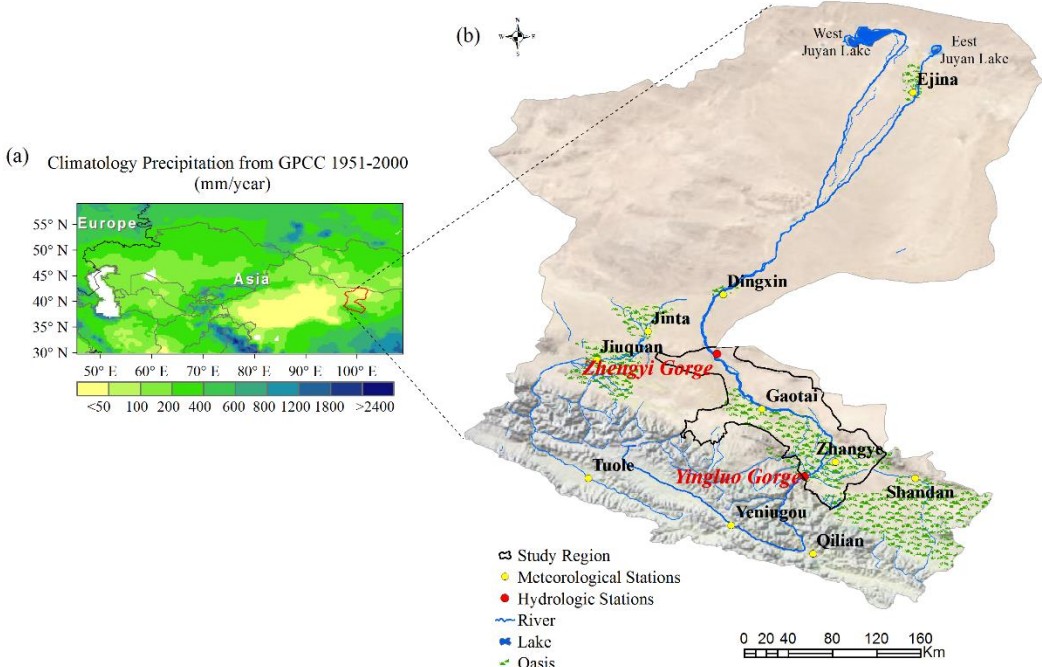


**Figure 1.** Details of the study area. (a) The location of Heihe River Basin and the mean annual precipitation (1951-2000). (b) Regional setting and the landscape of Heihe River Basin with location of meteorological and hydrologic stations.


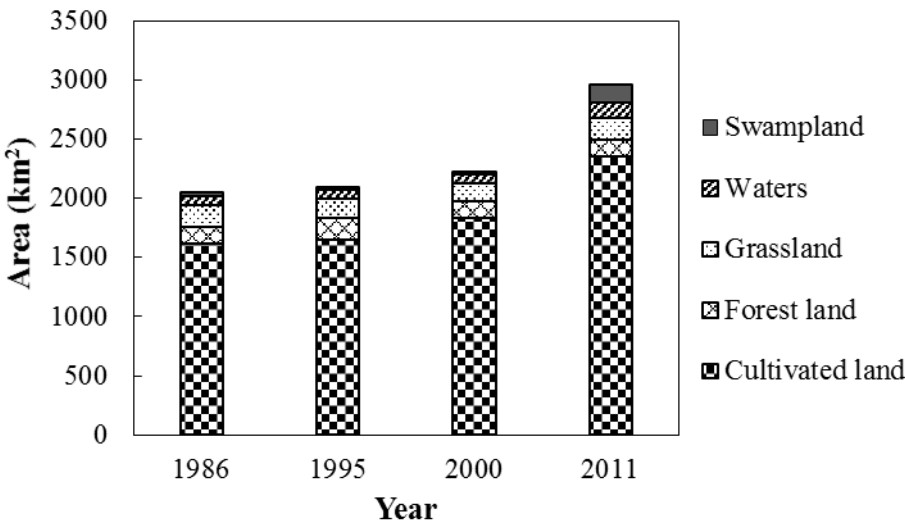

733 **Figure 2**. The areas of different land-use types in the oasis in the middle Heihe River Basin.


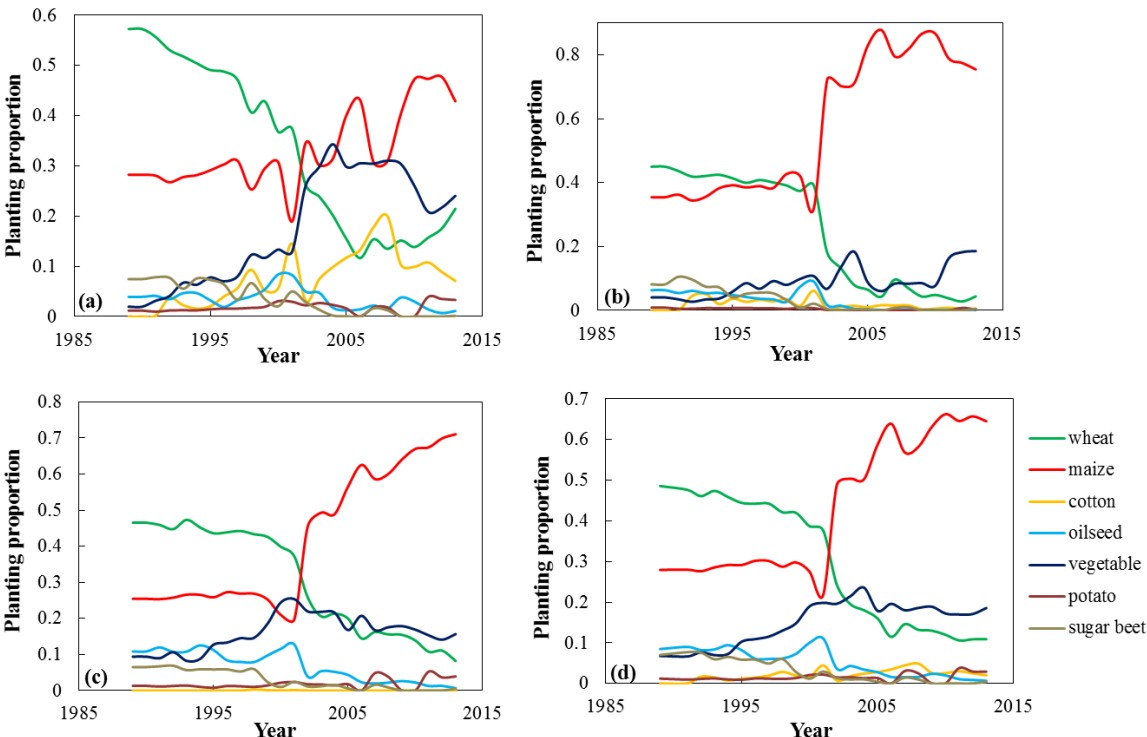

**Figure 3.** The planting proportion of the crops in (a) Gaotai county, (b) Linze county, (c) Ganzhou district, (d) the region including Gaotai, Linze counties and Ganzhou district in the middle Heihe River Basin.

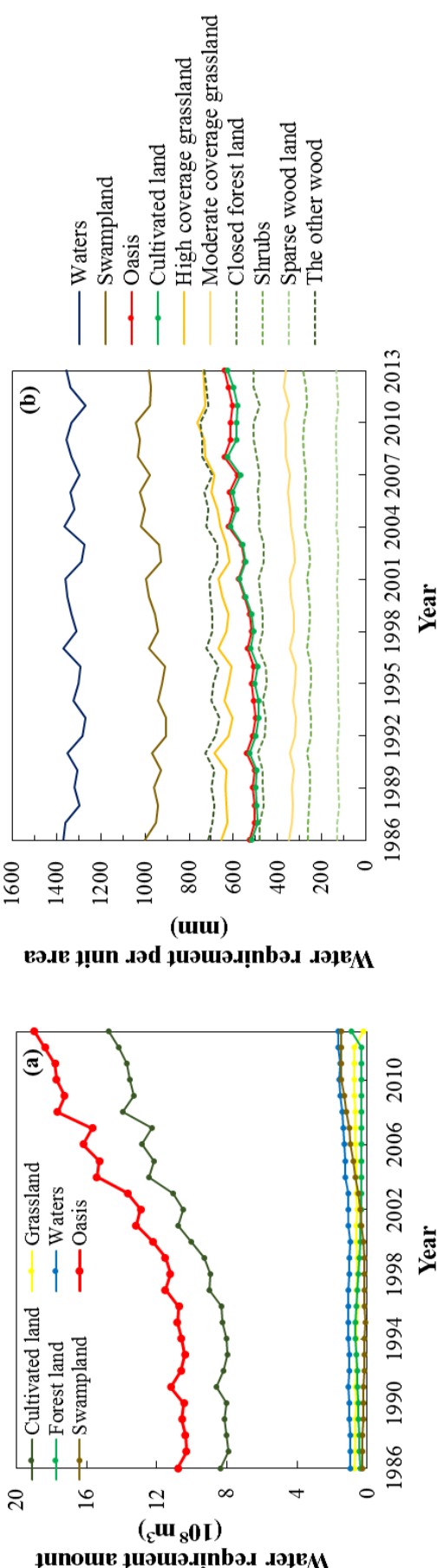

**Figure 4.** The water requirement amount (a) and water requirement per unit area (b) for different land-use types in the oasis of the middle Heihe River Basin from 1986 to 2013.

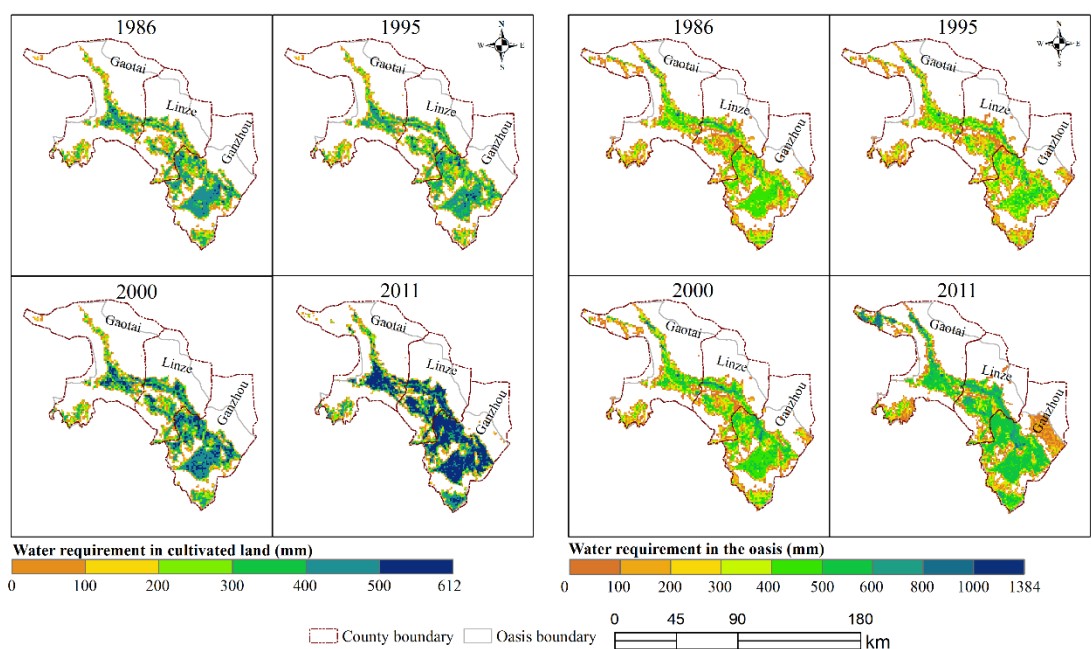

**Figure 5.** The spatial distribution of the water requirement in the cultivated land and oasis at the spatial resolution of 1 km in the middle Heihe River Basin in 1986, 1995, 2000, 2011.



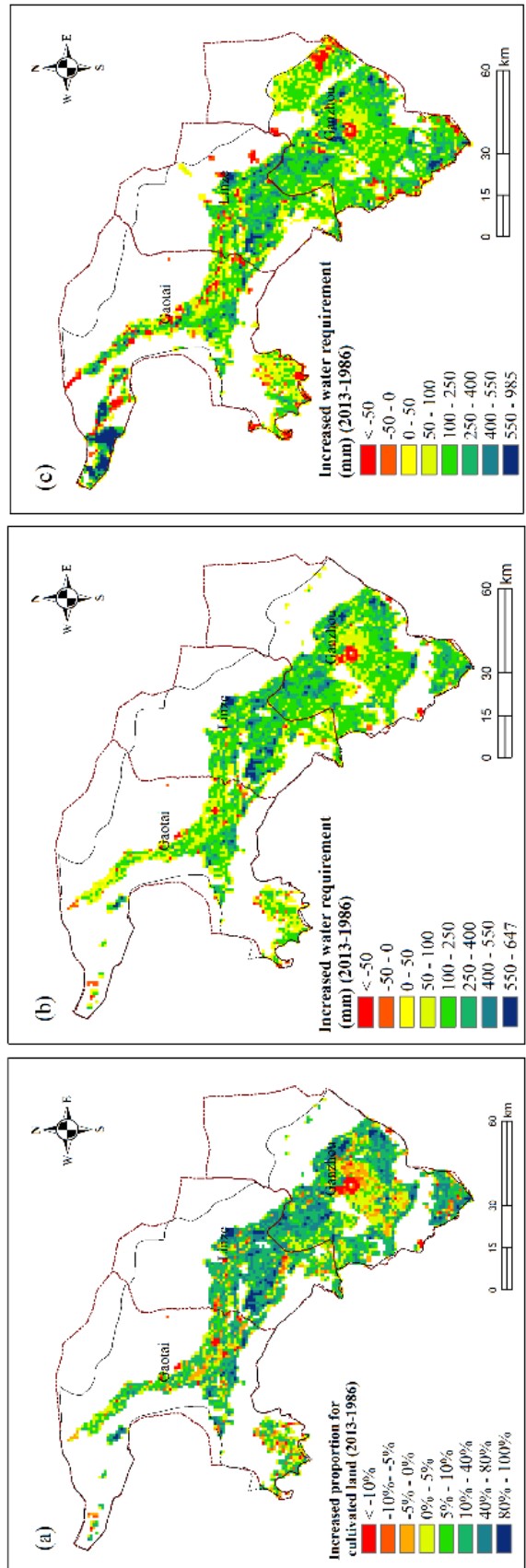

**Figure 6.** The difference of (a) the proportion of cultivated land, (b) water requirement in the cultivated land, and (c) water requirement in the oasis, at the spatial resolution of 1 km between 1986 and 2013 in the middle Heihe River Basin.

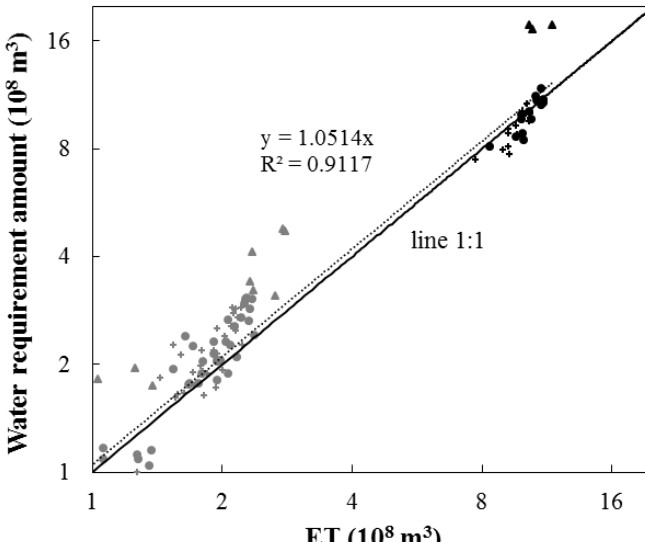

**Figure 7**. The comparison between the yearly and monthly (May, June, July) water requirement amounts
and ET data, which included the ET data estimated by ETWatch model from 2000 to 2013 for the
cultivated land and the oasis, and the ET data estimated by ETMonitor model from 2009 to 2011 for the
oasis.

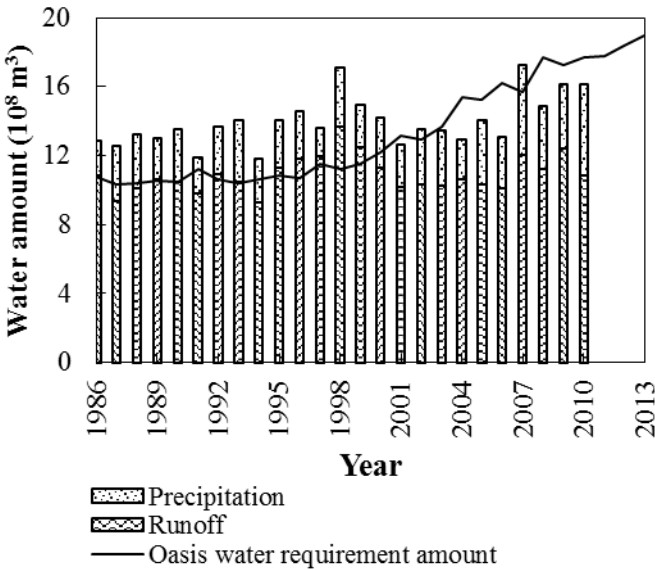


**Figure 8.** The water requirement amount and the surface water supply including the precipitation and runoff data for the oasis in the middle Heihe River Basin. The runoff data included the runoff from the mainstream of middle Heihe River, Shandan River and Liyuan River.



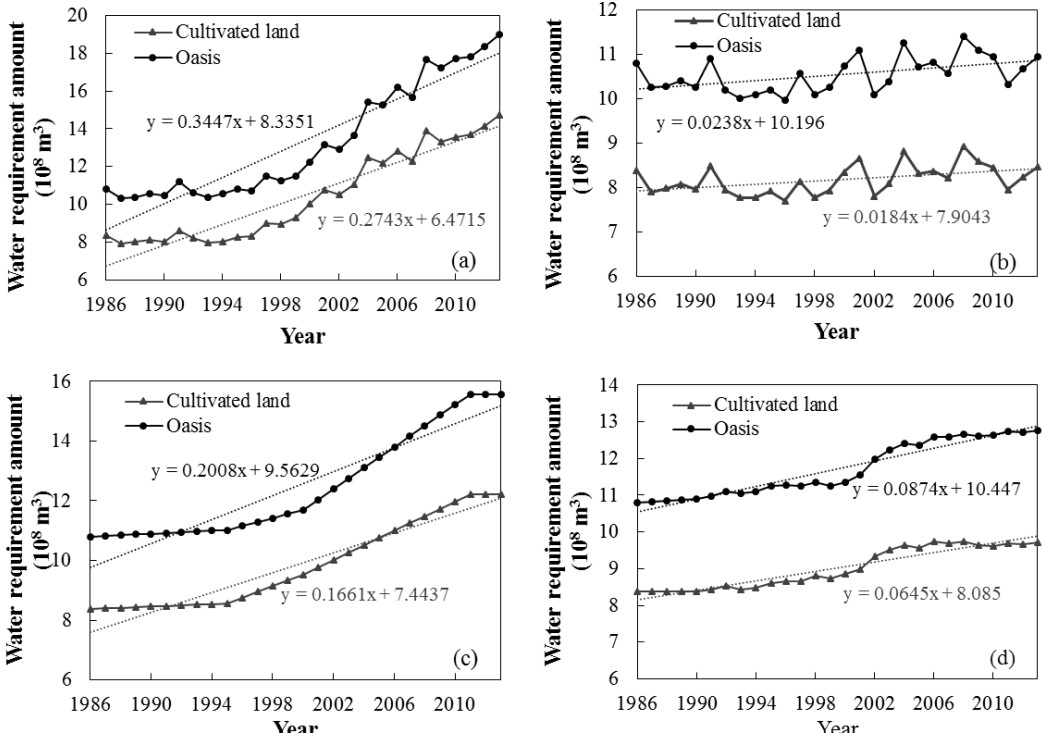

**Figure 9.** The long-term trend of the water requirement for the cultivated land and oasis under different
situations during 1986-2013. (a) Under a situation of climate change and human agricultural activities;
(b) under the situation that only the climate was changing, but the total area of the oasis (cultivated land)
and land structure were stable; (c) Under the situation that the total area of the oasis (cultivated land) was
changing, but the climate and land structure were stable; (d) Under the situation that the land structure
was changing, but the climate and the total area of the oasis (cultivated land) were stable.

Table 1. Crop coefficients of the different crops in different growth stages in the oasis of the middle Heihe River Basin.

| Crop | Development stage | | | |
|------|---------|------------|--------|------|
| | Initial | Developing | Middle | Late |
| Maize | 0.23 | 0.23-1.20 | 1.20 | 1.20-0.35 |
| Spring wheat | 0.23 | 0.23-1.16 | 1.16 | 1.16-0.40 |
| Cotton | 0.27 | 0.27-1.20 | 1.20 | 1.20-0.70 |
| Oilseed | 0.29 | 0.29-1.10 | 1.10 | 1.10-0.25 |
| Sugar beet | 0.34 | 0.34-1.21 | 1.21 | 1.21-0.70 |
| Potato | 0.27 | 0.27-1.15 | 1.15 | 1.15-0.75 |
| Vegetable | 0.60 | 0.60-1.10 | 1.10 | 1.10-0.90 |
| Orchard | 0.33 | 0.33-0.95 | 0.95 | 0.95-0.71 |
| Swampland | 1.00 | 1.00-1.20 | 1.20 | 1.20-1.00 |
| High coverage grassland | 0.20 | 0.20-1.04 | 1.04 | 1.04-0.44 |
| Moderate coverage grassland | 0.35 | 0.35-0.47 | 0.47 | 0.47-0.32 |


Table 2. Vegetation coefficient in different depths of groundwater level.

| Groundwater depth | 1 | 1.5 | 2 | 2.5 | 3 | 3.5 | 4 |
|---|---|---|---|---|---|---|---|
| Vegetation coefficient | 1.98 | 1.63 | 1.56 | 1.45 | 1.38 | 1.29 | 1.00 |


Table 3.Water balance items in the middle Heihe River Basin during 1986-2013.

| Average value (Unit: $10^8$ m$^3$) | | 1986-1989 | 1990-1999 | 2000-2013 |
|---|---|---|---|---|
| Water requirement | Agricultural vegetation | 8.32 | 8.84 | 12.61 |
| | Ecological vegetation | 2.19 | 2.07 | 3.26 |
| Runoff consumed in the middle Heihe River Basin | Mainstream of the middle Heihe River | 6.99 | 8.00 | 7.66[a] |
| | Shandan and Liyuan Rivers | 3.22 | 3.22 | 3.22 |
| Precipitation | Landing on the agricultural vegetation | 2.22 | 2.22 | 2.88 |
| | Landing on the ecological vegetation | 0.53 | 0.48 | 0.67 |
| Groundwater consumed in the middle Heihe River Basin | | 0.6[b] | 1.13 | 3.25[c] |

([a] the average value during 2000-2010, [c] the average value during 2000-2007; [b] the data referred to Yang and Wang (2005).)