# Peer review of "Quantify the impacts of climate change and human agricultural"

_Earth System Dynamics, 2017_

## Referee Comment (RC1) · Anonymous Referee #1 · 29 Sep 2017

The authors have presented us their nice work on calculating the water requirements over the selected basin (middle Heihe River). The changes in the water requirement per area or in total amount over different crops and land types are discussed very clearly. The contribution from different factors (climate, planting area changes and changes in land structure) was also analyzed with a linear method. Therefore this paper is a very good case study and it is well written as well. However, the study is not suitable for ESD due to the limited contribution to large scale researches.

Strong points: 1. The regional water requirement calculation is reliable based on the detailed data they have collected.

2. Attributing the increasing water requirement to different factors is a novel attempt to use the attribution methods.

I have no questions about the methods and results for the water requirement calculation or the attribution part.

Weak points: 1. I cannot see the points of comparing the simulation water requirement with the actual ET simulated by other models. Firstly, other models don't have the detailed data the authors have collected. Secondly, in an arid region, the actual ET is limited by the water available thus it can be largely different from the potential ET. Moreover, the reference data is too short to provide solid results, for example, in Figure 9, there are only three years and a few months samples.

2. A few questions on the water balance analysis. (1) In Table 3, what do you mean the "runoff for mainstream of the middle Heihe River"? Is it the river discharge flowing into the middle river basin (measured at Yingluo Gorge)? However, the value is not the same as you gave in the study area description. (2) In L106, you mentioned that there is more groundwater withdrawal in the basin recently. How much is the groundwater withdrawal compared to the surface water and to the water requirement? It will affect your water balance analysis in section 3.4 and Figure 10. (3) Is it possible to remove the water balance analysis since it is very close to the main topic.

3. The study area is very small with a catchment area of 8600 km2. It facilitates the water requirement calculation but it cannot reveal the general situation for large scales (i.e. basins, continents or globe), while it is the aim of journal ESD.
* * *

---

## Referee Comment (RC2) · Anonymous Referee #2 · 1 Oct 2017

This paper provides an assessment of the desert oasis water requirements during the past 3 decades in the middle Heihe River Basin. By using a GIS-based method, the proposed question of how and why the water requirement has changed during the period is answered. The regional water requirement is determined to be increased from $10.7 \times 108$ m3 in 1986 to $18.8 \times 108$ m3 in 2013. The extended planting area and the changes in land structure are confirmed to be the most important reasons for the increased water requirement, and climate changes does not significantly contribute to the trends. In general, the paper is clearly written, and the figures are informative and well presented. I have several minor comments for the authors to consider when revising the manuscript for publication (listed below line by line).

[Figure]

Detailed comments:

1. Part 1, Introduction: In this part, the author failed to review the existing literature thoroughly, and thus cannot naturally leads to a judgment whether the work is novel and significant when compared with previous published reports.

2. Line 197-203, Upon the Kc values, how did you dealing with the differences in Kc among different crop species? In my opinion, this value could be very different even in different varieties of the same crop types, especially when considering the species evolution in history.

3. Line 162-170, the validating data you used for comparison is also a GIS-base one, which were produced by using almost the same method and the same procedure. I guess a ground-based observation dataset may be more reasonable for the validating purpose in this paper.

4. Line 210-212, how do you treat the groundwater-dependent forest in the ecotone of desert-oasis system and the irrigation-based forest grids in agricultural land system?

5. Line 353, more details about ETMonitor model are needed here.

6. Line 440-444, I believe there are more than one field station in this region and meteorological observation and flux measurements were performed regularly over there.

7. Line 450-454, You mentioned the uncertainty brought into the calculation by crop coefficient, but how much this uncertainty it could be? Is that small enough to let you get sound conclusion?

---

## Author Comment (AC1) · 13 Oct 2017

We appreciate your comments and the effort you made to review the manuscript very much. Here are the responses for the weak points:

1. I cannot see the points of comparing the simulation water requirement with the actual ET simulated by other models. Firstly, other models don't have the detailed data the authors have collected. Secondly, in an arid region, the actual ET is limited by the water available thus it can be largely different from the potential ET. Moreover, the reference data is too short to provide solid results, for example, in Figure 9, there are only three years and a few months samples.

[Figure]

Response: Water requirement is defined as a theoretical value, and there are no available data to validate it directly. In most researches about water requirement, the validation part were omitted. But we wanted to validate it through an indirect way, so we used the actual ET data. Firstly, the spatial resolution of the actual ET data showed in Figure 8 is also 30 m as the resolution we used to calculate the water requirements. Secondly, the study area is dominated by the irrigated farmland which uses too much water from the Heihe River and causes serious ecological problems downstream. Though the irrigation cannot make the soil water conditions optimum all the time, the actual ET can also serve as a reference to some extent. Moreover, we used two kinds of reference data to validate the results. The data showed in Figure 9 is only one kind of the data, which can be taken as the supplementary data for the other kind data showed in Figure 8. We just want to use as much available ET data as possible.

2. A few questions on the water balance analysis. (1) In Table 3, what do you mean the "runoff for mainstream of the middle Heihe River"? Is it the river discharge flowing into the middle river basin (measured at Yingluo Gorge)? However, the value is not the same as you gave in the study area description. (2) In L106, you mentioned that there is more groundwater withdrawal in the basin recently. How much is the groundwater withdrawal compared to the surface water and to the water requirement? It will affect your water balance analysis in section 3.4 and Figure 10. (3) Is it possible to remove the water balance analysis since it is very close to the main topic.

Response: (1) The oasis is irrigated by the mainstream of Heihe River and some small tributaries like Liyuan River which have been separated from the mainstream. "The runoff for the mainstream of the middle Heihe River" means the river discharge of the mainstream of middle Heihe River flowing into the middle river basin. The middle Heihe River flows from Yingluo Gorge to Zhengyi Gorge, so the runoff for mainstream of the middle Heihe River is the difference of the river discharge between Yingluo Gorge and Zhengyi Gorge. We will make this part clear in the manuscript. (2) The mean annual groundwater withdrawal is about 0.33 billion m3. We will enrich the description

of groundwater withdrawal in section 2.1. And also we will add the groundwater withdrawal data into Figure 10 and analyze the water requirement with the surface water and groundwater withdrawal in section 3.4. (3) Water balance analysis can help researchers make clear the status of water supply and demand. We think it's meaningful to analyze the water balance.

3. The study area is very small with a catchment area of 8600 km2. It facilitates the water requirement calculation but it cannot reveal the general situation for large scales (i.e. basins, continents or globe), while it is the aim of journal ESD.

Response: Though the middle Heihe River Basin is not very large, but it's a very typical basin in arid regions, where river water originated from the high mountain area through snow/glacier melting and rainfall-runoff processes, consumed mainly by agriculture and human society in oases in middle reaches, and finally discharged and dispersed in the tail lake at the lower reach. All the rivers in arid region take the same hydrologic setting. Due to human over exploitation of water resources in the middle reach, i.e. oases, large amount of arid river basins are suffering severe ecological degradation in the world, such as Aral sea basin, Lake Chad basin, Tarim basin, etc. Because of the typicality and not too large to collect data, Heihe river basin was chosen as the target basin for a key national research programme on ecohydrology and integrated basin water management by the Natural Scientific Foundation of China in 2012. And the programme is still going on, this paper is totally supported by an key project of this national research programme. Even though Heihe river basin is not a large basin, but insight of the nexus relationship between agriculture, hydrology, and ecology is considered to be significant and able to reveal the general characteristics of the arid basins. As for the facts of Heihe River basin, it is the second largest inland river in China. More than 90 percent of the population and arable land in the Heihe River Basin were concentrated in the midstream, where the most water in the whole basin was consumed. Additionally, the middle river basin is the grain base for the Northwestern China. To restore the ecological environment and mediate the competition for water
between agriculture development and environmental health, the Chinese government have invested 1.49 billion Yuan in the middle Heihe River Basin. The importance and typicality of the study region make the study valuable to the research for the arid regions in the world.

---

## Author Comment (AC2) · 19 Oct 2017

Thank you very much for the comments and the effort you made to review the manuscript. The comments are very valuable to the manuscript. According to the comments, we have made clear of the runoff data, and analyzed the water balance part considering the groundwater withdrawal. Unfortunately, we only got the groundwater withdrawal data from 1990 to 2007, so we didn't put the groundwater withdrawal data into Figure 10. But we put the data into Table 3 because we can only get an average value of the groundwater withdrawal in 1980s besides the groundwater withdrawal during 1990-2007. Please refer to the supplement to see the revised manuscript.

[Figure]

Please also note the supplement to this comment:
https://www.earth-syst-dynam-discuss.net/esd-2017-75/esd-2017-75-AC2-
supplement.zip

---

## Author Response (AR1)

**Final response**

The authors gratefully thank to the editor and the anonymous reviewers for the valuable comments on our manuscript which drive us to improve the manuscript greatly. We made a major revision of our manuscript, the title of the manuscript was changed to make it more fit to the content of our study as "Quantify the impacts of climate change and human agricultural activities on oasis water requirements in an arid region: A case study of the Heihe River Basin, China". And the introduction part, study area part, spatial variations of the oasis, and discussion part were rewrote to make our manuscript more clear and scientific. The figures in the manuscript have been rearranged. The comments and questions were addressed point by point as below. The text quoted from the revised manuscript is shown in red.

**Response to Referee #1**

We thank the anonymous reviewer for the constructive comments on the manuscript and the effort made to review the manuscript very much. Here are the point-by-point responses for the weak points.

1. I cannot see the points of comparing the simulation water requirement with the actual ET simulated by other models. Firstly, other models don't have the detailed data the authors have collected. Secondly, in an arid region, the actual ET is limited by the water available thus it can be largely different from the potential ET. Moreover, the reference data is too short to provide solid results, for example, in Figure 9, there are only three years and a few months samples.

   **Response**: Thank you for the comments. In this study, water requirement is defined as a theoretical value, and there are no available data to validate it directly. In most researches about water requirement, the validation part were omitted. But we wanted to validate it through an indirect way, so we used the actual ET data. Firstly, the spatial resolution of the actual ET data estimated by ETWatch showed in the original Figure 8 was also 30 m as the same with the resolution we used to calculate the water requirements. The resolution was stated in part 2.2.3, and could also be found in the part of "Validation of the oasis water requirements validation". Secondly, the study area was dominated by the irrigated farmland which used too much water from the Heihe River and caused serious ecological problems downstream. Though the irrigation could not make the soil water conditions optimum all the time, the actual ET can also serve as a reference to some extent. Moreover, we used two kinds of reference data to validate the results. The data showed in the original Figure 9 was only one kind of the data, which could be taken as the supplementary data for the other kind data showed in the original Figure 8. We wanted to use as much available ET data as possible. In the revised manuscript, we rearranged the figures, the original Figure 8 and Figure 9 were combined into one new figure named Figure 7.

2. A few questions on the water balance analysis. (1) In Table 3, what do you mean the "runoff for mainstream of the middle Heihe River"? Is it the river discharge flowing into the middle river basin (measured at Yingluo Gorge)? However, the value is not the same as you gave in the study area description. (2) In L106, you mentioned that there is more groundwater withdrawal in the basin recently. How much is the groundwater withdrawal compared to the surface water and to the water requirement? It will affect your water balance analysis in section 3.4 and Figure 10. (3) Is it possible to remove the water balance analysis since it is very close to the main topic.

**Response:** We appreciate the valuable comments very much.

(1) The oasis is irrigated by the mainstream of middle Heihe River and some small tributaries like Liyuan River and Shandan River which have been separated from the mainstream. "The runoff for the mainstream of the middle Heihe River" means the runoff consumed in the middle Heihe River Basin from the mainstream of the middle Heihe River. The middle Heihe River flows from Yingluo Gorge to Zhengyi Gorge, so the runoff discharge flowing into the middle river basin is the difference of the river discharge between Yingluo Gorge and Zhengyi Gorge. We have changed the "runoff" to "runoff consumed in the middle Heihe River Basin" in Table 3 and clarified this point in the part of "Water balance in the middle Heihe River Basin".

(2) We have enriched the description of groundwater withdrawal in section 2.1 as below:

"According to the groundwater withdrawal data (1990-2007) of the irrigation districts in the middle Heihe River Basin (Wu et al., 2014; Zheng, 2014), which were downloaded from the Cold and Arid Regions Science Data Center at Lanzhou (http://westdc.westgis.ac.cn), only $1.13\times10^8$ m$^3$ of groundwater was pumped on average in 1990s, but the amount increased to $3.25\times10^8$ m$^3$ on average during 2000-2007."

Unfortunately, we only got the groundwater withdrawal data from 1990 to 2007, so we didn't put the groundwater withdrawal data into the original Figure 10. But we put the data into Table 3 because we only got an average value of the groundwater withdrawal in 1980s besides the groundwater withdrawal during 1990-2007. And we also revised some of the water balance analysis as below:

"Compared with the available water resources in 1980s, precipitation had remained little changed in 1990s. But with the increase of water requirement in 1990s, the runoff consumed in the middle Heihe River Basin had an obvious rise and more groundwater was pumped for irrigation (Fig. 9; Table 3). Ignoring the industrial and domestic water taken from the middle Heihe River, the surface water supply seemed to be sufficient to the water requirement in the oasis in 1980s and 1990s. While entering the 20th century, the area of arable land increased fast, and high water-requiring crops (maize and vegetable) had gradually replaced the low water-requiring crop (wheat) since 2001 (Fig. 3). Therefore, the water requirement increased a lot in 2000s. With the implementation of EWCP, the available surface water from middle Heihe River decreased in 2000s. Surface water cannot meet the water requirement any more, causing more exploiting of groundwater (Table 3). The middle Heihe River Basin is in severe water shortage of water resources. To reduce the contradiction of water supply and requirement, the land use including the crop structure in the middle Heihe River Basin should be carefully planned."

(3) Water balance analysis can help researchers make clear the status of water supply and demand. We think it's meaningful to analyze the water balance.

3. The study area is very small with a catchment area of 8600 km2. It facilitates the water requirement calculation but it cannot reveal the general situation for large scales (i.e. basins, continents or globe), while it is the aim of journal ESD.

**Response**: Though the middle Heihe River Basin is not very large, but it's a very typical basin in arid regions, where river water originated from the high mountain area through snow/glacier melting and rainfall-runoff processes, consumed mainly by agriculture and human society in oases in middle reaches, and finally discharged and dispersed in the tail lake at the lower reach. All the rivers in arid region take the same hydrologic setting. Due to human over exploitation of water resources in the middle reach, i.e. oases, large amount of arid river basins are suffering severe ecological degradation in the world, such as Aral Sea basin, Lake Chad basin, Tarim basin, etc. Because of the typicality and not too large to collect data, Heihe river basin was chosen as the target basin for a key national research programme on ecohydrology and integrated basin water management by the Natural Scientific Foundation of China in 2012. And the programme is still going on, this paper is totally supported by a key project of this national research programme. Even though Heihe river basin is not a large basin, but insight of the nexus relationship between agriculture, hydrology, and ecology is considered to be significant and able to reveal the general characteristics of the arid basins. As for the facts of Heihe River basin, it is the second largest inland river in China. More than 90 % of the population and arable land in the Heihe River Basin were concentrated in the midstream, where the most water in the whole basin was consumed. Additionally, the middle river basin is the grain base for the Northwestern China. To restore the ecological environment and mediate the competition for water between agriculture development and environmental health, the Chinese government have invested 1.49 billion Yuan in the middle Heihe River Basin. The importance and typicality of the study region make the study valuable to the research for the arid regions in the world. We have rewrote the introduction part to make clear of the significance of our study.

**Response to Referee #2**

The authors gratefully thank to the valuable comments on our manuscript which drives us to improve the manuscript greatly. According to the comments, we have rewrote the parts of Introduction, Study area, and Discussion. The comments and questions were addressed point by point as below:

1.  Part 1, Introduction: In this part, the author failed to review the existing literature thoroughly, and thus cannot naturally leads to a judgment whether the work is novel and significant when compared with previous published reports.

    **Response:** We appreciate this comment very much. We have reviewed much more literature and rewrote the introduction part. We hope the revised introduction have shown the novel and significant of our work naturally. The introduction part was rewrote as below:

[revised manuscript text omitted]

2. Line 197-203, Upon the Kc values, how did you dealing with the differences in Kc among different crop species? In my opinion, this value could be very different even in different varieties of the same crop types, especially when considering the species evolution in history.

**Response**: The differences in Kc among different crop species were determined by the references Duan et al. (2004) and FAO (Allen et al., 1998). There are different Kc values for different crop species in different growth stages. These were stated in part 2.3.1 and the Kc values for different crop species were shown in Table 1. But we didn't distinguish the crop coefficients among different varieties of the same crop type.

We do agree with you that the Kc values could be very different even in different varieties of the same crop type, especially when considering the species evolution in history. But it's difficult to get the crop coefficients for every specific crop variety because there are too many varieties. Besides, the water requirement is not only related to the crop coefficient, but also related to the crop growth period. Many factors influencing the crop coefficient are also have an effect on the growth stages. Limited by the available data, the Kc values in the same growth stage for different varieties of the same crop type were the same. So we discussed this topic in the discussion part.

3. Line 162-170, the validating data you used for comparison is also a GIS-base one, which were produced by using almost the same method and the same procedure. I guess a ground-based observation dataset may be more reasonable for the validating purpose in this paper.

**Response**: Thank you for your suggestion very much. When we worked on the validation part, we also thought to validate the results by using the ground-based observation dataset. But we found that the available ground-based ET datasets by eddy covariance system were not enough to validate the results, which only included the data from around June, 2012 to December, 2013 during the study period. And many daily data were incomplete. So we didn't use the ground-based observation dataset to validate the results.

4. Line 210-212, how do you treat the groundwater-dependent forest in the ecotone of desert-oasis system and the irrigation-based forest grids in agricultural land system?

**Response**: The irrigation-based forest in agricultural land system is always shelter forest for farmland which usually belongs to the closed forest land in the land use dataset. The natural vegetation in the ecotone of desert-oasis system is always sparse wood or shrubs due to the lack of precipitation. Water requirement is a theoretical value which is calculated assuming that the water is always suitable. So we didn't distinguish the forest land in the ecotone of desert-oasis system from that in the agricultural land system, and just treated the forest land as closed forest land, sparse wood land and shrubs as the land use dataset showed.

5. Line 353, more details about ETMonitor model are needed here.

**Response**: ETMonitor model is a hybrid remotely sensed ETa estimation model developed by Hu G., and Jia, L. (2015). Please refer to the reference:

Hu G., and Jia, L.: Monitoring of evapotranspiration in a semi-arid inland river basin by combining microwave and optical remote sensing observations, Remote Sensing, 7(3), 3056-3087, 2015.

In part 2.2.3, there were introductions of the validation data which included the data estimated by ETMonitor model. We have added more information about this model in this part as below:

"The other was the daily ET datasets (2009-2011) of the Heihe River Basin at 1 km spatial resolution (Cui and Jia, 2014; Jia et al., 2013) estimated by ETMonitor, which is a hybrid remotely sensed actual ET estimation model developed by Hu and Jia (2015)."

6. Line 440-444, I believe there are more than one field station in this region and meteorological observation and flux measurements were performed regularly over there.

**Response**: Yes, there are some field stations in this region and meteorological observation and flux measurements that were performed regularly. But the measured ET are actual ET. We don't know whether the irrigation can make the soil water keep optimum for the crops all the time during the whole growth period. And the measurements began from around June, 2012. The available data were limited. We have enriched and made this point clear in the discussion part as below:

"Crop water requirement is the ET from disease-free, well-fertilized crops under optimum soil water conditions and achieving full production. There is no available observed crop water requirement to validate the results. Only actual ET data can be obtained. There are 18 field stations in the oasis that all located in Ganzhou district in the middle Heihe River Basin for conducting meteorological observation and flux measurements from around June, 2012. But due to the incomplete daily data and short time series, we used the ET datasets provided by Cold and Arid Regions Science Data Center at Lanzhou (http://westdc.westgis.ac.cn) to validate the results."

7. Line 450-454, You mentioned the uncertainty brought into the calculation by crop coefficient, but how much this uncertainty it could be? Is that small enough to let you get sound conclusion?

**Response**: Thank you for the valuable comment. We enriched the discussion part about the uncertainty of crop coefficients to make the uncertainty be clear as below:

[revised manuscript text omitted]

- Yearly data of the oasis compared with ETWatch data

- Monthly data of the oasis compared with ETWatch data

+ Yearly data of the cultivated land compared with ETWatch data

+ Monthly data of the cultivated land compared with ETWatch data

▲ Yearly data of the oasis compared with ETMonitor data

▲ Monthly data of the oasis compared with ETMonitor data

**Figure 7**. The comparison between the yearly and monthly (May, June, July) water requirement amounts
and ET data, which included the ET data estimated by ETWatch model from 2000 to 2013 for the
cultivated land and the oasis, and the ET data estimated by ETMonitor model from 2009 to 2011 for the
oasis.

[Figure]

**Figure 8.** The comparison between the yearly water requirement amount and ET from 2000 to 2013 (a); monthly water requirement amount and ET in May, June, and July from 2000 to 2013 (b) for the cultivated land and the oasis.

[Figure]

**Figure 9.** The comparison between the water requirements and ET of the oasis, including the yearly data from 2009 to 2011 and the monthly data in May, June and July.

[Figure]

[Figure]

**Figure 8.** The water requirement amount and the surface water supply including the precipitation and runoff data for the oasis in the middle Heihe River Basin. The runoff data included the runoff from the mainstream of middle Heihe River, Shandan River and Liyuan River.

[Figure]

**Figure 9**. The long-term trend of the water requirement for the cultivated land and oasis under different
situations during 1986-2013.
(a) Under a situation of climate change and human agricultural activities; (b) under the situation that only the climate was changing, but the total area of the oasis (cultivated land) and  land structure were stable ; (c) Under the situation that the total area of the oasis (cultivated land) was changing, but the climate and land structure were stable ; (d) Under the situation that  the land structure was changing, but the climate and the total area of the oasis (cultivated land) were stable .

Table 1. Crop coefficients of the different crops in different growth stages in the oasis of the middle Heihe River Basin.

| Crop | Development stage | | | |
|---|---|---|---|---|
| | Initial | Developing | Middle | Late |
| Maize | 0.23 | 0.23-1.20 | 1.20 | 1.20-0.35 |
| Spring wheat | 0.23 | 0.23-1.16 | 1.16 | 1.16-0.40 |
| Cotton | 0.27 | 0.27-1.20 | 1.20 | 1.20-0.70 |
| Oilseed | 0.29 | 0.29-1.10 | 1.10 | 1.10-0.25 |
| Sugar beet | 0.34 | 0.34-1.21 | 1.21 | 1.21-0.70 |
| Potato | 0.27 | 0.27-1.15 | 1.15 | 1.15-0.75 |
| Vegetable | 0.60 | 0.60-1.10 | 1.10 | 1.10-0.90 |
| Orchard | 0.33 | 0.33-0.95 | 0.95 | 0.95-0.71 |
| Swampland | 1.00 | 1.00-1.20 | 1.20 | 1.20-1.00 |
| High coverage grassland | 0.20 | 0.20-1.04 | 1.04 | 1.04-0.44 |
| Moderate coverage grassland | 0.35 | 0.35-0.47 | 0.47 | 0.47-0.32 |

Table 2. Vegetation coefficient in different depths of groundwater level.

| Groundwater depth | 1 | 1.5 | 2 | 2.5 | 3 | 3.5 | 4 |
|---|---|---|---|---|---|---|---|
| Vegetation coefficient | 1.98 | 1.63 | 1.56 | 1.45 | 1.38 | 1.29 | 1.00 |

Table 3.Water balance items in the middle Heihe River Basin during 1986-2013.

| Average value (Unit: $10^8$ m$^3$) | | 1986-1989 | 1990-1999 | 2000-2013 |
|---|---|---|---|---|
| Water requirement | Agricultural vegetation | 8.32 | 8.84 | 12.61 |
| | Ecological vegetation | 2.19 | 2.07 | 3.26 |
| Runoff consumed in the middle Heihe River Basin | Mainstream of the middle Heihe River | 6.99 | 8.00 | 7.66[a] |
| | Shandan and Liyuan Rivers | 3.22 | 3.22 | 3.22 |
| Precipitation | Landing on the agricultural vegetation | 2.22 | 2.22 | 2.88 |
| | Landing on the ecological vegetation | 0.53 | 0.48 | 0.67 |
| Groundwater consumed in the middle Heihe River Basin | | 0.6[b] | 1.13 | 3.25[c] |

([a] the average value during 2000-2010, [c] the average value during 2000-2007; [b] the data referred to Yang and Wang (2005).)